# On the unreasonable effectiveness of feature propagation in learning on graphs with missing node features

## Abstract

While Graph Neural Networks (GNNs) have recently become the *de facto* standard for modeling relational data, they impose a strong assumption on the availability of the node or edge features of the graph. In many real-world applications, however, features are only partially available; for example, in social networks, age and gender are available only for a small subset of users. We present a general approach for handling missing features in graph machine learning applications that is based on minimization of the Dirichlet energy and leads to a diffusion-type differential equation on the graph. The discretization of this equation produces a simple, fast and scalable algorithm which we call Feature Propagation. We experimentally show that the proposed approach outperforms previous methods on seven common node-classification benchmarks and can withstand surprisingly high rates of missing features: on average we observe only around 4% relative accuracy drop when 99% of the features are missing. Moreover, it takes only 10 seconds to run on a graph with $\sim$2.5M nodes and $\sim$123M edges on a single GPU.

## 1 Introduction

Graph Neural Networks (GNNs) (Gori et al., 2005; Scarselli et al., 2008; Kipf & Welling, 2017; Gilmer et al., 2017a; Veličković et al., 2018; Bronstein et al., 2017) have been successful on a broad range of problems and in a variety of fields (Duvenaud et al., 2015; Ying et al., 2018; Zitnik et al., 2018; Gainza et al., 2020; Sanchez-Gonzalez et al., 2020; Shlomi et al., 2020; Derrow-Pinion et al., 2021). GNNs typically operate by a message-passing mechanism (Battaglia et al., 2018; Gilmer et al., 2017b), where at each layer, nodes send their feature representations ("messages") to their neighbors. The feature representation of each node is initialized to their original features, and is updated by repeatedly aggregating incoming messages from neighbors. Being able to combine the topological information with feature information is what distinguishes GNNs from other purely topological learning approaches such as random walks (Perozzi et al., 2014; Grover & Leskovec, 2016) or label propagation (Zhu & Ghahramani, 2002), and arguably what leads to their success.

GNN models typically assume a fully observed feature matrix, where rows represent nodes and columns feature channels. However, in real-world scenarios, each feature is often only observed for a subset of the nodes. For example, demographic information can be available for only a small subset of social network users, while content features are generally only present for the most active users. In a co-purchase network, not all products may have a full description associated with them. With the rising awareness around digital privacy, data is increasingly available only upon explicit user consent. In all the above cases, the feature matrix contains missing values and most existing GNN models cannot be directly applied.

While classic imputation methods (Liu et al., 2020; Yoon et al., 2018; Kingma & Welling, 2014) can be used to fill the missing values of the feature matrix, they are unaware of the underlying graph structure. Graph Signal Processing, a field attempting to generalize classical Fourier analysis to graphs, offers several methods that reconstruct signals on graphs (Narang et al., 2013). However, they do not scale beyond graphs with a few thousand nodes, making them infeasible for practical applications. More recently, SAT (Chen et al., 2020), GCNMF (Taguchi et al., 2021) and PaGNN (Jiang & Zhang, 2021) have been proposed to adapt GNNs to the case of missing features.

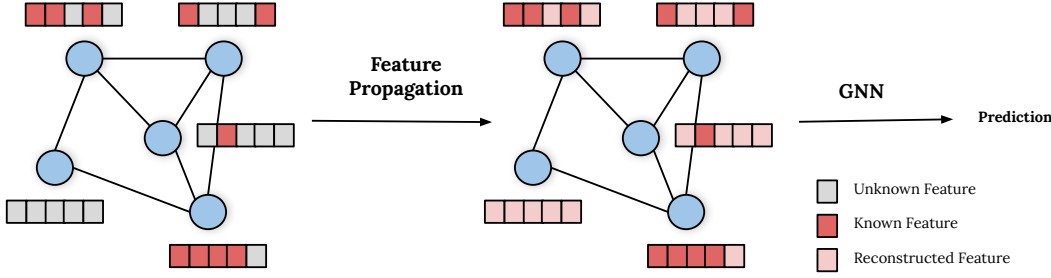

Figure 1: A diagram illustrating our Feature Propagation framework. On the left, a graph with missing node features. In the initial reconstruction step, Feature Propagation reconstructs the missing features by iteratively diffusing the known features in the graph. Subsequently, the graph and the reconstructed node features are fed into a downstream GNN model, which then produces a prediction.

However, they are not evaluated at high missing features rates ($> 90\%$), which occur in many real-world scenarios, and where we find them to suffer. Moreover, they are unable to scale to graphs with more than a few hundred thousand nodes. At the time of writing, PaGNN is the state-of-the-art method for node classification with missing features.

**Contributions** We present a general approach for handling missing node features in graph machine learning tasks. The framework consists of an initial diffusion-based feature reconstruction step followed by a downstream GNN. The reconstruction step is based on Dirichlet energy minimization, which leads to a diffusion-type differential equation on the graph. Discretization of this differential equation leads to a very simple, fast, and scalable iterative algorithm which we call Feature Propagation (FP). FP outperforms state-of-the-art methods on six standard node-classification benchmarks and presents the following advantages:

- **Theoretically Motivated**: FP emerges naturally as the gradient flow minimizing the Dirichlet energy and can be interpreted as a diffusion equation on the graph with known features used as boundary conditions.

- **Robust to high rates of missing features**: FP can withstand surprisingly high rates of missing features. In our experiment, we observe on average around 4% relative accuracy drop when up to 99% of the features are missing. In comparison, GCNMF and PaGNN have an average drop of 53.33% and 21.25% respectively.

- **Generic**: FP can be combined with any GNN model to solve the downstream task; in contrast, GCNMF and PaGNN are specific GCN-type models.

- **Fast and Scalable**: FP takes only around 10 seconds for the reconstruction step on OGBN-Products (a graph with $\sim$2.5M nodes and $\sim$123M edges) on a single GPU. GCNMF and PaGNN run out-of-memory on this dataset.

## 2 PRELIMINARIES

Let $G = (V, E)$ be an undirected graph with $n \times n$ adjacency matrix $\mathbf{A}$ and a node feature vector[1] $\mathbf{x} \in \mathbb{R}^n$.

The *graph Laplacian* is an $n \times n$ positive semi-definite matrix $\mathbf{\Delta} = \mathbf{I} - \tilde{\mathbf{A}}$, where $\tilde{\mathbf{A}} = \mathbf{D}^{-\frac{1}{2}} \mathbf{A} \mathbf{D}^{-\frac{1}{2}}$ is the normalized adjacency matrix and $\mathbf{D} = \mathrm{diag}(\sum_j a_{1j}, \ldots, \sum_j a_{nj})$ is the diagonal degree matrix.

---

[1]For convenience, we assume scalar node features. Our derivations apply straightforwardly to the case of $d$-dimensional features represented as an $n \times d$ matrix $\mathbf{X}$.

Denote by $V_k \subseteq V$ the set of nodes on which the features are *known*, and by $V_u = V_k^c = V \setminus V_k$ the *unknown* ones. We further assume the ordering of the nodes such that we can write

$$\mathbf{x} = \begin{bmatrix} \mathbf{x}_k \\ \mathbf{x}_u \end{bmatrix} \qquad \mathbf{A} = \begin{bmatrix} \mathbf{A}_{kk} & \mathbf{A}_{ku} \\ \mathbf{A}_{uk} & \mathbf{A}_{uu} \end{bmatrix} \qquad \mathbf{\Delta} = \begin{bmatrix} \mathbf{\Delta}_{kk} & \mathbf{\Delta}_{ku} \\ \mathbf{\Delta}_{uk} & \mathbf{\Delta}_{uu} \end{bmatrix}.$$

Because the graph is undirected, $\mathbf{A}$ is symmetric and thus $\mathbf{A}_{ku}^\top = \mathbf{A}_{uk}$ and $\mathbf{\Delta}_{ku}^\top = \mathbf{\Delta}_{uk}$. We will tacitly assume this in the following discussion.

**Graph feature interpolation** is the problem of reconstructing the unknown features $\mathbf{x}_u$ given the graph structure $G$ and the known features $\mathbf{x}_k$. The interpolation task requires some prior on the behavior of the features of the graph, which can be expressed in the form of an energy function $\ell(\mathbf{x}, G)$. The most common assumption is feature *homophily* (i.e., that the features of every node are similar to those of the neighbours), quantified using a criterion of *smoothness* such as the Dirichlet energy. Since in many cases the behavior of the features is not known, the energy can possibly be learned from the data.

**Learning on a graph with missing features** is a transductive learning problem (typically node-wise classification or regression using some GNN architecture) where the structure of the graph $G$ is known while the labels and node features are only partially known on the subsets $V_l$ and $V_k$ of nodes, respectively (that might be different and even disjoint). Specifically, we try to learn a function $\mathbf{f}(\mathbf{x}_k, G)$ such that $f_i \approx y_i$ for $i \in V_l$. Learning with missing features can be done by a pre-processing step of graph signal interpolation (reconstructing an estimate $\tilde{\mathbf{x}}$ of the full feature vector $\mathbf{x}$ from $\mathbf{x}_k$) independent of the learning task, followed by the learning task of $\mathbf{f}(\tilde{\mathbf{x}}, G)$ on the inferred fully-featured graph. In some settings, we are not interested in recovering the features *per se*, but rather ensuring that the output of the *function* $\mathbf{f}$ on these features is correct – arguably a more 'forgiving' setting.

## 3 FEATURE PROPAGATION

We assume to be given $\mathbf{x}_k$ and attempt to find the missing node features $\mathbf{x}_u$ by means of interpolation that minimizes some energy $\ell(\mathbf{x}, G)$. In particular, we consider the *Dirichlet energy* $\ell(\mathbf{x}, G) = \frac{1}{2}\mathbf{x}^\top \mathbf{\Delta} \mathbf{x} = \frac{1}{2} \sum_{ij} \tilde{a}_{ij}(x_i - x_j)^2$, where $\tilde{a}_{ij}$ are the individual entries of the normalized adjacency $\tilde{\mathbf{A}}$. The Dirichlet energy is widely used as a smoothness criterion for functions defined on the nodes of the graph and thus promotes homophily. Functions minimizing the Dirichlet energy are called *harmonic*; without boundary conditions, it is minimized by a constant function.

While the Dirichlet energy is convex and it is possible to derive its minimizer in a closed-form, as shown below, the computational complexity makes it unfeasible for graphs with many nodes with missing features. Instead, we consider the associated *gradient flow* $\dot{\mathbf{x}}(t) = -\nabla\ell(\mathbf{x}(t))$ as a differential equation with boundary condition $\mathbf{x}_k(t) = \mathbf{x}_k$ whose solution at the missing nodes, $\mathbf{x}_u = \lim_{t\to\infty} \mathbf{x}_u(t)$, provides the desired interpolation.

**Gradient flow.** For the Dirichlet energy, $\nabla_{\mathbf{x}}\ell = \mathbf{\Delta}\mathbf{x}$ and the gradient flow takes the form of the standard isotropic heat diffusion equation on the graph,

$$\dot{\mathbf{x}}(t) = -\mathbf{\Delta}\mathbf{x}(t) \qquad \text{(IC) } \mathbf{x}(0) = \begin{bmatrix} \mathbf{x}_k \\ \mathbf{x}_u(0) \end{bmatrix} \qquad \text{(BC) } \mathbf{x}_k(t) = \mathbf{x}_k$$

where IC and BC stand for initial conditions and boundary conditions respectively. By incorporating the boundary conditions, we can compactly express the diffusion equation as

$$\begin{bmatrix} \dot{\mathbf{x}}_k(t) \\ \dot{\mathbf{x}}_u(t) \end{bmatrix} = - \begin{bmatrix} \mathbf{0} & \mathbf{0} \\ \mathbf{\Delta}_{uk} & \mathbf{\Delta}_{uu} \end{bmatrix} \begin{bmatrix} \mathbf{x}_k \\ \mathbf{x}_u(t) \end{bmatrix} = - \begin{bmatrix} \mathbf{0} \\ \mathbf{\Delta}_{uk}\mathbf{x}_k + \mathbf{\Delta}_{uu}\mathbf{x}_u(t) \end{bmatrix}. \tag{1}$$

As expected, the gradient flow of the observed features is $\mathbf{0}$, given that they do not change during the diffusion.

The evolution of the missing features can be regarded as a heat diffusion equation with a constant heat source $\mathbf{\Delta}_{uk}\mathbf{x}_k$ coming from the boundary (known) nodes. Since the graph Laplacian matrix is

positive semi-definite, the Dirichlet energy $\ell$ is convex. Its global minimizer is given by the solution to the closed-form equation $\nabla_{\mathbf{x}_u}\ell = \mathbf{0}$ and by rearranging the final $|V_u|$ rows of Equation 1 we get the solution $\mathbf{x}_u = -\boldsymbol{\Delta}_{uu}^{-1}\boldsymbol{\Delta}_{ku}^{\top}\mathbf{x}_k$. This solution always exists as $\boldsymbol{\Delta}_{uu}$ is non-singular, by virtue of the following:

**Proposition 1** (The sub-Laplacian matrix of an undirected connected graph is invertible). *Take any undirected, connected graph with adjacency matrix $\mathbf{A} \in \{0,1\}^{n \times n}$, and its Laplacian $\boldsymbol{\Delta} = \mathbf{I} - \mathbf{D}^{-1/2}\mathbf{A}\mathbf{D}^{-1/2}$, with $\mathbf{D}$ being the degree matrix of $\mathbf{A}$. Then, for any principle sub-matrix $\mathbf{L}_u \in \mathbb{R}^{b \times b}$ of the Laplacian, where $1 \leq b < n$, $\mathbf{L}_u$ is invertible.*

Proof: See Appendix A. Also, while the proposition assumes that the graph is connected, our analysis and method generalize straightforwardly in the cases of disconnected graph as we can simply apply Feature Propagation to each connected component independently.

However, solving a system of linear equations is computationally expensive (incurring $\mathcal{O}(|V_u|^3)$ complexity for matrix inversion) and thus intractable for anything but only small graphs.

**Iterative scheme.** As an alternative, we can discretize the diffusion equation (1) and solve it by an iterative numerical scheme. Approximating the temporal derivative as forward difference with the time variable $t$ discretized using a fixed step ($t = hk$ for step size $h > 0$ and $k = 1, 2, \ldots$), we obtain the *explicit Euler scheme:*

$$\mathbf{x}^{(k+1)} = \mathbf{x}^{(k)} - h \begin{bmatrix} \mathbf{0} & \mathbf{0} \\ \boldsymbol{\Delta}_{uk} & \boldsymbol{\Delta}_{uu} \end{bmatrix} \mathbf{x}^{(k)} = \left( \mathbf{I} - \begin{bmatrix} \mathbf{0} & \mathbf{0} \\ h\boldsymbol{\Delta}_{uk} & h\boldsymbol{\Delta}_{uu} \end{bmatrix} \right) \mathbf{x}^{(k)} = \begin{bmatrix} \mathbf{I} & \mathbf{0} \\ -h\boldsymbol{\Delta}_{uk} & \mathbf{I} - h\boldsymbol{\Delta}_{uu} \end{bmatrix} \mathbf{x}^{(k)}$$

For the special case of $h = 1$, we can use the following observation

$$\tilde{\mathbf{A}} = \mathbf{I} - \boldsymbol{\Delta} = \begin{bmatrix} \mathbf{I} & \mathbf{0} \\ \mathbf{0} & \mathbf{I} \end{bmatrix} - \begin{bmatrix} \boldsymbol{\Delta}_{kk} & \boldsymbol{\Delta}_{ku} \\ \boldsymbol{\Delta}_{uk} & \boldsymbol{\Delta}_{uu} \end{bmatrix} = \begin{bmatrix} \mathbf{I} - \boldsymbol{\Delta}_{kk} & -\boldsymbol{\Delta}_{ku} \\ -\boldsymbol{\Delta}_{uk} & \mathbf{I} - \boldsymbol{\Delta}_{uu} \end{bmatrix},$$

to write the iteration formula as

$$\mathbf{x}^{(k+1)} = \begin{bmatrix} \mathbf{I} & \mathbf{0} \\ \tilde{\mathbf{A}}_{uk} & \tilde{\mathbf{A}}_{uu} \end{bmatrix} \mathbf{x}^{(k)}. \tag{2}$$

The Euler scheme is the gradient descent of the Dirichlet energy. Thus, applying the scheme decreases the Dirichlet energy and results in the features becoming increasingly smooth. Iteration (2) can be interpreted as successive low-pass filtering. Figure 2 depicts the magnitude of the graph Fourier coefficients of the original and reconstructed features on the Cora dataset, indicating that the higher the rate of missing features, the stronger the low-pass filtering effect.

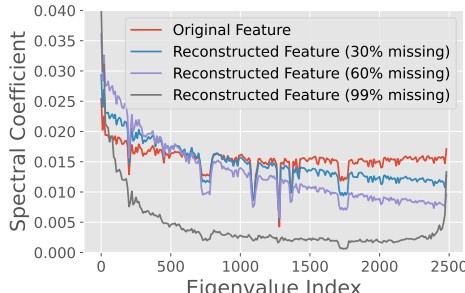

The following results shows that the iterative scheme with $h = 1$ always converges and its steady state is equal to the closed form solution. Importantly, the solution does not depend on the initial values $\mathbf{x}_u^{(0)}$ given to the unknown features.

**Proposition 2.** *Take any undirected and connected graph with adjacency matrix $\mathbf{A} \in \{0,1\}^{n \times n}$, and normalised Adjacency $\tilde{\mathbf{A}} = \mathbf{D}^{-1/2}\mathbf{A}\mathbf{D}^{-1/2}$, with $\mathbf{D}$ being the degree matrix of $A$. Let $\mathbf{X} \in \mathbf{R}^{n \times d} = \mathbf{X}^{(0)} \in \mathbf{R}^{n \times d}$ be a feature matrix and define the following recursive relation*

Figure 2: Graph Fourier transform magnitudes of the original Cora features (red) and those reconstructed by FP for varying rates of missing rates (we take the average over feature channels). Since FP minimizes the Dirichlet energy, it can be interpreted as a low-pass filter, which is stronger for a higher rate of missing features.

$$\mathbf{X}^{(k)} = \begin{bmatrix} \mathbf{I} & \mathbf{0} \\ \tilde{\mathbf{A}}_{uk} & \tilde{\mathbf{A}}_{uu} \end{bmatrix} \mathbf{X}^{(k-1)}.$$

*Then this recursion converges and the steady state is given to be*

$$\lim_{n \to \infty} \mathbf{X}^{(n)} = \begin{bmatrix} \mathbf{X}_k \\ -\boldsymbol{\Delta}_{kk}^{-1}\tilde{\mathbf{A}}_{uk}\mathbf{X}_k \end{bmatrix}.$$

Proof: See Appendix A.

**Feature Propagation Algorithm.** Equation 2 provides an extremely simple and scalable iterative algorithm to reconstruct the missing features, which we refer to as *Feature Propagation* (FP). While $\mathbf{x}_u$ can be initialized to any value, we initialize $\mathbf{x}_u$ to zero and find 40 iterations to be enough to provide convergence for all datasets we experimented on. At each iteration, the diffusion occurs from the nodes with known features to the nodes with unknown features as well as among the nodes with unknown features.

---

**Algorithm 1** Feature Propagation

1: $\mathbf{y} \leftarrow \mathbf{x}$
2: **while** $\mathbf{x}$ has not converged **do**
3: $\quad \mathbf{x} \leftarrow \tilde{\mathbf{A}}\mathbf{x}$ $\qquad \triangleright$ Propagate features
4: $\quad \mathbf{x}_k \leftarrow \mathbf{y}_k$ $\qquad \triangleright$ Reset known features
5: **end while**

---

It is worth noting that the proposed algorithm bears some similarity with Label Propagation (Zhu & Ghahramani, 2002) (LP), which predicts a class for each node by propagating the known labels in the graph. Differently from our setting of diffusion of continuous node features, they deal with discrete label classes directly, resulting in a different diffusion operator. However, the key difference between them lies in how they are used. LP is used to directly perform node classification, taking into account only the graph structure and being unaware of node features. On the other hand, FP is used to reconstruct missing features, which are then fed into a downstream GNN classifier. FP allows a GNN model to effectively combine features and graph structures, even when most of the features are missing. Our experiments show that FP+GNN always outperforms LP, even in cases of extremely high rates of missing features, suggesting the effectiveness of FP. Also, the derived scheme is a special case of Neural Graph PDEs, and in turn it is also related to the iterative scheme presented in Zhou & Schölkopf (2004).

**Extension to Vector-Valued Features.** The method extends to vector-valued features by simply replacing the feature vector $\mathbf{x}$ with a $n \times d$ feature matrix $\mathbf{X}$ in Algorithm 1, where $d$ is the number of features. Multiplying the diffusion matrix $\mathbf{A}$ by the feature matrix $\mathbf{X}$ diffuses each feature channel independently.

**Learning.** One significant advantage of FP is that it can be easily combined with any graph learning model to generate predictions for the downstream task. Moreover, FP is not aimed at merely reconstructing the node features, and instead by only reconstructing the lower frequency components of the signal, it is by design very well suited to be combined with GNNs, which are known to mainly leverage these lower frequency components (Wu et al., 2019). Our approach is generic and can be used for any graph-related task for missing features, such as node classification, link prediction and graph classification. In this paper, we focus on node classification.

## 4 RELATED WORK

**Matrix completion.** Several optimization-based approaches (Candès & Recht, 2009; Hu et al., 2008) as well as learning-based approaches (Liu et al., 2020; Yoon et al., 2018; Kingma & Welling, 2014) have been proposed to solve the matrix completion problem. However, they are unaware of the underlying graph structure. Graph matrix completion (Kalofolias et al., 2014; van den Berg et al., 2017; Monti et al., 2017; Rao et al., 2015) extends the above approaches to make use of an underlying graph. Similarly, Graph Signal Processing offers several methods to interpolate signals on graphs. Narang et al. (2013) prove the necessary conditions for a graph signal to be recovered perfectly, and provide a corresponding algorithm. However, due to the optimisation problems involved, most above approaches are too computationally intensive and cannot scale to graphs with more than $\sim$1,000 nodes. Moreover, the goal of all above approaches is to reconstruct the missing entries of the matrix, rather than solving a downstream task.

**Extending GNNs to missing node features.** SAT (Chen et al., 2020) consists of a Transformer-like model for feature reconstruction and a GNN model to solve the downstream task. GC-NMF (Taguchi et al., 2021) adapts GCN (Kipf & Welling, 2017) to the case of missing node features by representing the missing data with a Gaussian mixture model. PaGNN (Jiang & Zhang, 2021) is a GCN-like model which uses a partial message-passing scheme to only propagate observed features.

While showing a reasonable performance for low rates of missing features, these methods suffer in regimes of high rates of missing features, and do not scale to large graphs.

**Other related GNN works.** Several papers investigate how to augment GNNs when no node features are available (Cui et al., 2021), as well as investigating the performance of GNNs with random features (Sato et al., 2021; Abboud et al., 2021). Dirichlet energy minimization has been widely used as a regularizer in several graph-related tasks (Zhu et al., 2003; Zhou & Schölkopf, 2004; Weston et al., 2008). Discretizion of continuous diffusion on graphs has already been explored in Chamberlain et al. (2021) and Xhonneux et al. (2020).

## 5 EXPERIMENTS AND DISCUSSION

**Datasets.** We evaluate on the task of node classification on several benchmark datasets: Cora, Citeseer and PubMed (Sen et al., 2008), Amazon-Computers, Amazon-Photo (Shchur et al., 2018) and OGBN-Arxiv (Hu et al., 2020). To test the scalability of our method, we also test it on OGBN-Products (2,449,029 nodes, 123,718,280 edges). We report dataset statistics in table 3 (Appendix).

**Baselines.** We compare to Label Propagation (Zhu & Ghahramani, 2002), a strong feature-agnostic baseline which only makes use of the graph structure by propagating labels on the graph. We additionally compare to feature-imputation methods that are graph-agnostic, such as setting the missing features to 0 (Zero), a random value from a standard Gaussian (Random), or the global mean of that feature over the graph (Global Mean) [2]. We also compare to a simple graph-based imputation baseline, which sets a missing feature to the mean (of that same feature) over the neighbors of a node (Neighbor Mean). We additionally experiment with MGCNN (Monti et al., 2017), a geometric graph completion method which learns how to reconstruct the missing features by making use of the observed features and the graph structure. For all the above imputation and matrix completion baselines, as well as for our Feature Propagation, we experiment with both GCN (Kipf & Welling, 2017) and GraphSage with mean aggregator (Hamilton et al., 2017) as downstream GNNs. We also compare to recently state-of-the-art methods for learning in the missing features setting (GC-NMF (Taguchi et al., 2021) and PaGNN (Jiang & Zhang, 2021)). For GCNMF we use the publicly available code.[3] We could not find publicly available code for PaGNN so use our own implementation for this comparison. We do not compare to other commonly used imputation based methods such as VAE (Kingma & Welling, 2014) or GAIN (Yoon et al., 2018), nor to the Transformer-based method SAT (Chen et al., 2020), as they have previously been shown to consistently underperform GCNMF and PaGNN (Taguchi et al., 2021; Jiang & Zhang, 2021).

**Experimental Setup.** We report the mean and standard error of the test accuracy, computed over 10 runs, in all experiments. Each run has a different train/validation/test split (apart from OGBN datasets where we use the provided splits) and mask of missing features[4]. The splits are generated at random by assigning 20 nodes per class to the training set, 1500 nodes in total to the validation set and the rest to the test set, similar to Klicpera et al. (2019). For a fair comparison, we use the same standard hyperparameters for all methods across all experiments. We train using the Adam (Kingma & Ba, 2015) optimizer with a learning rate of 0.005 for a maximum of 10000 epochs, combined with early stopping with a patience of 200. Downstream GNN models (as well as GCNMF and PaGNN) use 2 layers with a hidden dimension of 64 and a dropout rate of 0.5 for all datasets, apart from OGBN datasets where 3 layers and a hidden dimension of 256 are used. For OGBN-Arxiv we also employ the Jumping Knowledge scheme (Xu et al., 2018) with max aggregation. Feature Propagation uses 40 iterations to diffuse the features. We want to emphasize that we did not perform any hyperparameter tuning, and FP proved to perform consistently with any reasonable choice of hyperparameters. We use neighbor sampling (Hamilton et al., 2017) when training on OGBN-Products. All experiments are conducted on an AWS p3.16xlarge machine with 8 NVIDIA V100 GPUs[5].

---

[2]If a feature is not observed for any of the node's neighbors, we set it to zero.

[3]https://github.com/marblet/GCNmf

[4]Each entry of the feature matrix is independently missing with a probability equal to the missing rate.

[5]Each V100 GPU has 16GB of memory.

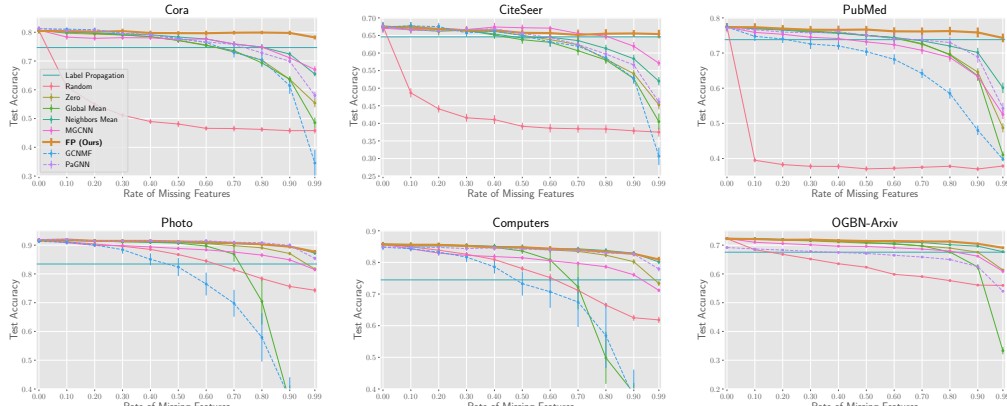

Figure 3: Test accuracy for varying rate of missing features on six common node-classification benchmarks. For methods that require a downstream GNNs, a 2-layer GCN (Kipf & Welling, 2017) is used. On OGBN-Arxiv, GCNMF goes out-of-memory and is not reported.

| Dataset | Full Features | 50.0% Missing | 90.0% Missing | 99.0% Missing |
|---|---|---|---|---|
| Cora | 80.39% | 79.70%(-0.86%) | 79.77%(-0.77%) | 78.22%(-2.70%) |
| CiteSeer | 67.48% | 65.74%(-2.57%) | 65.57%(-2.82%) | 65.40%(-3.08%) |
| PubMed | 77.36% | 76.68%(-0.89%) | 75.85%(-1.96%) | 74.29%(-3.97%) |
| Photo | 91.73% | 91.29%(-0.48%) | 89.48%(-2.46%) | 87.73%(-4.36%) |
| Computers | 85.65% | 84.77%(-1.04%) | 82.71%(-3.43%) | 80.94%(-5.51%) |
| OGBN-Arxiv | 72.22% | 71.42%(-1.10%) | 70.47%(-2.43%) | 69.09%(-4.33%) |
| OGBN-Products | 78.70% | 77.16%(-1.96%) | 75.94%(-3.51%) | 74.94%(-4.78%) |
| Average | 79.08% | 78.11%(-1.27%) | 77.11%(-2.48%) | 75.80%(-4.10%) |

Table 1: Performance of Feature Propagation (combined with a GCN model) for 50%, 90% and 99% of missing features, and relative drop compared to the performance of the same model when all features are present. On average, our method loses only 2.50% of relative accuracy with 90% of missing features, and 4.12% with 99% of missing features.

| Dataset | GCNMF | PaGNN | Label Propagation | FP (Ours) |
|---|---|---|---|---|
| Cora | 34.54±2.07 | 58.03±0.57 | 74.68±0.36 | **78.22**±0.32 |
| CiteSeer | 30.65±1.12 | 46.02±0.58 | 64.60±0.40 | **65.40**±0.54 |
| PubMed | 39.80±0.25 | 54.25±0.70 | 73.81±0.56 | **74.29**±0.55 |
| Photo | 29.64±2.78 | 85.41±0.28 | 83.45±0.94 | **87.73**±0.27 |
| Computers | 30.74±1.95 | 77.91±0.33 | 74.48±0.61 | **80.94**±0.37 |
| OGBN-Arxiv | OOM | 53.98±0.08 | 67.56±0.00 | **69.09**±0.06 |
| OGBN-Products | OOM | OOM | 74.42±0.00 | **74.94**±0.07 |

Table 2: Performance of GCNMF, PaGNN and FP(+GCN) with 99% of features missing, as well as Label Propagation (which is feature-agnostic). GCNMF and PaGNN perform respectively 58.33% and 21.25% worse in terms of relative accuracy in this scenario compared to when all the features are present. In comparison, FP has only a 4.12% drop.

**Node Classification Results.** Figure 3 shows the results for different rates of missing features (x-axis), when using GCN as a downstream GNN (results with GraphSAGE are reported in Figure 6 of the Appendix). FP matches or outperforms other methods in all scenarios. Both GCNMF and PaGNN are consistently outperformed by the simple Neighbor Mean baseline. This is not completely unexpected, as Neighbor Mean can be seen as a first-order approximation of Feature Propa-

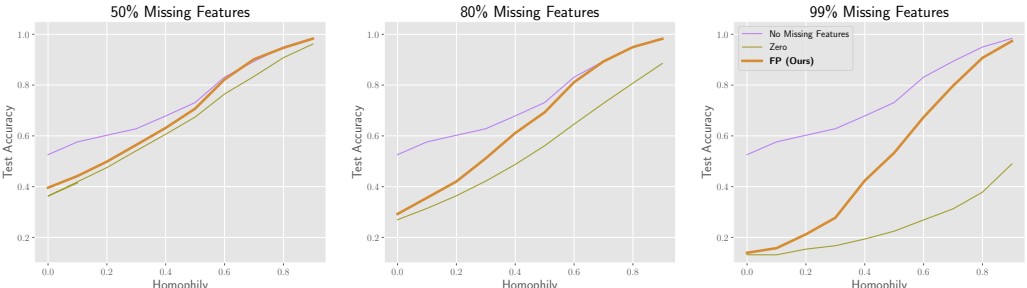

Figure 5: Test accuracy on the synthetic datasets from Abu-El-Haija et al. (2019) with different levels of homophily. We use GraphSage as downstream model as it is preferable to GCN on low homophily data (Zhu et al., 2020).

gation, where only one step of propagation is performed (and with a slightly different normalization of the diffusion operator). We elaborate on the relation between Neighbor Mean and Feature Propagation as well as on the results of the other baselines in Section A.4 of the Appendix. Interestingly, most methods perform extremely well up to 50% of missing features, suggesting that in general node features are redundant, as replacing half of them with zeroa (*Zero* baseline) has little effect on the performance. The gap between methods opens up from around 60% of missing features, and is particularly large for extremely high rates of missing features (90% or 99%): FP is the only feature-aware method which is robust to these high rates on all datasets (see Table 2). Moreover, FP outperforms Label Propagation on all datasets, even in the extreme case of 99% missing features. On some datasets, such as Cora, Photo, and Computers, the gap is especially significant. We conclude that reconstructing the missing features using FP is indeed useful for the downstream task. We highlight the surprising results that, on average, FP with 99% missing features performs only 4.12% worse (in relative accuracy terms) than the same GNN model used with no missing features, compared to 58.33% and 21.25% for GCNMF and PaGNN respectively.

**Run-time analysis.** Feature Propagation scales to extremely large graphs, as it only consists of repeated sparse-to-dense matrix multiplications. Moreover, it can be regarded as a pre-processing step, and performed only once, separately from training. In Figure 4 we compare the run-time to complete the training of the model for FP, PaGNN and GCNMF. The time for FP includes both the feature propagation step to reconstruct

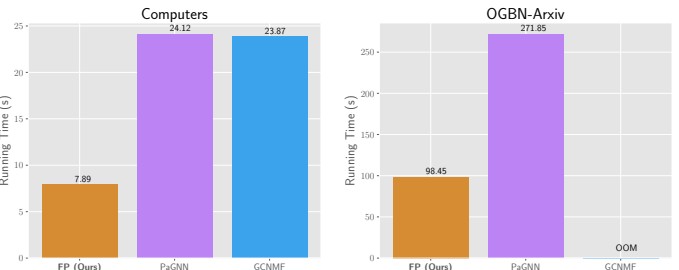

Figure 4: Run-time (in seconds) of FP, PaGNN and GCNMF. FP is 3x faster than both other methods. GCNMF goes out-of-memory (OOM) on OGBN-Arxiv.

the missing features, as well as training of a downstream GCN model. FP is around 3x faster than PaGNN and GCNMF. The propagation step of FP takes only a fraction of the total running time, and the vast majority of the time is spent in training of the donwstream model. The feature propagation step takes only ∼0.6s for Computers, ∼0.8s for OGBN-Arxiv and ∼10.5s for OGBN-Products using a single GPU. Both PaGNN and GCNMF go out-of-memory on OGBN-Products.

**When does Feature Propagation work?** Since FP can be interpreted as a low-pass filter that smoothes the features on the graph, we expect it to be suitable in the case of homophilic graph data (where neighbors tend to have similar attributes), and, conversely, to suffer in scenarios of low homophily. To verify this, we experiment on the synthetic dataset from Abu-El-Haija et al. (2019), which consists of 10 graphs with different levels of homophily. Figure 5 confirms our hypothesis: when the homophily is high, Feature Propagation performs similarly to the case when all the features are known. As the homophily decreases, the gap between the two widens to become extremely large

in cases of zero homophily. In such scenarios, FP is only slightly better than setting the missing features to zero (Zero baseline). This observation calls for a different kind of non-homogeneous diffusion dependent on the features that can potentially be made learnable for low-homophily data. We leave this as future work.

## 6 CONCLUSION

We have introduced a novel approach for handling missing node features in graph-learning tasks. Our Feature Propagation model can be directly derived from energy minimization, and can be implemented as an efficient iterative algorithm where the features are multiplied by a diffusion matrix, before resetting the known features to their original value. Experiments on a number of datasets suggest that FP can reconstruct the missing features in a way that is useful for the downstream task, even when $99\%$ of the features are missing. FP outperforms recently proposed methods by a significant margin on common benchmarks, while also being extremely scalable.

**Limitations.** While our method is designed for homophilic graphs, a more general learnable diffusion could be adopted to perform well in low homophily scenarios, as discussed in Section 5. Feature Propagation is designed for graphs with only one node and edge type, however it could be extended to heterogenous graphs by having separate diffusions for different types of edges and nodes. Finally, Feature Propagation treats feature channels independently. To account for dependencies, diffusion with channel mixing should be used.

**Reproducibility Statement.** The datasets used in this paper are publicly available and can be obtained from either the Pytorch-Geometric library (Fey & Lenssen, 2019) or Open Graph Benchmark (Hu et al., 2020). In the supplementary material we provide our code, together with a README file explaining in detail how to reproduce the results presented in the paper. All hyperparameters and their values are listed in section 5.

**Ethical Statement.** Our work is aimed at improving the performance of Graph Neural Networks. While we believe that nothing in our work raises specific ethical concerns, the recent broad adoption of GNNs in industrial applications opens the possibility to the misuse of such methods with potentially detrimental societal impact.

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

## A  APPENDIX

### A.1  CLOSED-FORM SOLUTION FOR HARMONIC INTERPOLATION

Given the *Dirichlet energy* $\ell(\mathbf{x}, G) = \frac{1}{2}\mathbf{x}^\top \mathbf{\Delta x}$, we want to solve for missing features $\mathbf{x}_u = argmin_{\mathbf{x}_u}\ell$, leading to the optimality condition $\nabla_{\mathbf{x}_u}\ell = \mathbf{0}$. From Eq. 1 we find $\nabla_{\mathbf{x_u}}\ell = \mathbf{0}$ to be the solution of $\mathbf{\Delta}_{uk}\mathbf{x}_k + \mathbf{\Delta}_{uu}\mathbf{x}_u = \mathbf{0}$. The unique solution to this system of linear equations is $\mathbf{x}_u = -\mathbf{\Delta}_{uu}^{-1}\mathbf{\Delta}_{uk}\mathbf{x}_k$. We show this solution always exists by proving $\mathbf{\Delta}_{uu}$ is non-singular (Proposition 1). The proof of this result follows from the following Lemma.

**Lemma 1.** *Take any undirected and connected graph with adjacency matrix* $\mathbf{A} \in \{0, 1\}^{n \times n}$, *and normalised Adjacency* $\tilde{\mathbf{A}} = \mathbf{D}^{-1/2}\mathbf{A}\mathbf{D}^{-1/2}$, *with* $\mathbf{D}$ *being the degree matrix of A. Let* $\tilde{\mathbf{A}}_{uu}$ *be the bottom right submatrix of* $\tilde{\mathbf{A}}$ *where* $1 \le b < n$. *Then* $\rho(\tilde{\mathbf{A}}_{uu}) < 1$ *where* $\rho(\cdot)$ *denotes spectral radius.*

*Proof.* Define

$$\tilde{\mathbf{A}}_{up} = \begin{bmatrix} \mathbf{0}_u & \mathbf{0}_{uk} \\ \mathbf{0}_{ku} & \tilde{\mathbf{A}}_{uu} \end{bmatrix},$$

to be the matrix equal to $\tilde{\mathbf{A}}_{uu}$ in the lower right $b \times b$ sub-matrix and padded with zero entries elsewhere. Clearly $\tilde{\mathbf{A}}_{up} \leq \tilde{\mathbf{A}}$ elementwise and $\tilde{\mathbf{A}}_{up} \neq \tilde{\mathbf{A}}$. Furthermore, $\tilde{\mathbf{A}}_{up} + \tilde{\mathbf{A}}$ represents an adjacency matrix of some strongly connected graph and is therefore irreducible (Berman & Plemmons, 1994, Theorem 2.2.7). These observations allow us to deduce that $\rho(\tilde{\mathbf{A}}_{up}) < \rho(\tilde{\mathbf{A}})$ (Berman & Plemmons, 1994, Corollary 2.1.5). Note that $\rho(\tilde{\mathbf{A}}_{up}) = \rho(\tilde{\mathbf{A}}_{uu})$ as $\tilde{\mathbf{A}}_{up}$ and $\tilde{\mathbf{A}}_{uu}$ share the same non-zero eigenvalues. Furthermore, $\rho(\tilde{\mathbf{A}}) \leq 1$ as we can write $\tilde{\mathbf{A}} = \mathbf{I} - \mathbf{\Delta}$ and $\mathbf{\Delta}$ is known to have eigenvalues in the range $[0, 2]$ (Chung, 1997). Combining these inequalities gives the result $\rho(\tilde{\mathbf{A}}_{uu}) = \rho(\tilde{\mathbf{A}}_{up}) < \rho(\tilde{\mathbf{A}}) \leq 1$. □

**Proposition 1** (The sub-Laplacian matrix of a undirected connected graph is invertible). *Take any undirected, connected graph with adjacency matrix $\mathbf{A} \in \{0, 1\}^{n \times n}$, and its Laplacian $\mathbf{\Delta} = \mathbf{I} - \mathbf{D}^{-1/2}\mathbf{A}\mathbf{D}^{-1/2}$, with $\mathbf{D}$ being the degree matrix of $\mathbf{A}$. Then, for any principle sub-matrix $\mathbf{L}_u \in \mathbb{R}^{b \times b}$ of the Laplacian, where $1 \leq b < n$, $L_u$ is invertible.*

*Proof.* To prove $\mathbf{\Delta}_{uu}$ is non-singular it is enough to show $0$ is not an eigenvalue. Note that $\mathbf{\Delta}_{uu} = \mathbf{I} - \tilde{\mathbf{A}}_{uu}$ so $0$ is not an eigenvalue if and only if $\tilde{\mathbf{A}}_{uu}$ does not have an eigenvalue equal to $1$, which follows from Lemma 1. □

### A.2 CLOSED-FORM SOLUTION FOR THE EULER SCHEME

**Proposition 2.** *Take any undirected and connected graph with adjacency matrix $\mathbf{A} \in \{0, 1\}^{n \times n}$, and normalised Adjacency $\tilde{\mathbf{A}} = \mathbf{D}^{-1/2}\mathbf{A}\mathbf{D}^{-1/2}$, with $\mathbf{D}$ being the degree matrix of $A$. Let $\mathbf{X} \in \mathbf{R}^{n \times d} = \mathbf{X}^{(0)} \in \mathbf{R}^{n \times d}$ be a feature matrix and define the following recursive relation*

$$\mathbf{X}^{(k)} = \begin{bmatrix} \mathbf{I}_l & \mathbf{0}_{ku} \\ \tilde{\mathbf{A}}_{uk} & \tilde{\mathbf{A}}_{uu} \end{bmatrix} \mathbf{X}^{(k-1)}.$$

*Then this recursion converges and the steady state is given to be*

$$\lim_{n \to \infty} \mathbf{X}^{(n)} = \begin{bmatrix} \mathbf{X}_k \\ -\mathbf{\Delta}_{kk}^{-1}\tilde{\mathbf{A}}_{uk}\mathbf{X}_k \end{bmatrix}.$$

*Proof.* The recursive relation can be written in the following form

$$\begin{bmatrix} \mathbf{X}_k^{(k)} \\ \mathbf{X}_u^{(k)} \end{bmatrix} = \begin{bmatrix} \mathbf{I}_l & \mathbf{0}_{ku} \\ \tilde{\mathbf{A}}_{uk} & \tilde{\mathbf{A}}_{uu} \end{bmatrix} \begin{bmatrix} \mathbf{X}_k^{(k-1)} \\ \mathbf{X}_u^{(k-1)} \end{bmatrix} = \begin{bmatrix} \mathbf{X}_k^{(k-1)} \\ \tilde{\mathbf{A}}_{uk}\mathbf{X}_k^{(k-1)} + \tilde{\mathbf{A}}_{uu}\mathbf{X}_u^{(k-1)} \end{bmatrix}.$$

The first $l$ rows remain the same so we can write $\mathbf{X}_k^{(k)} = \mathbf{X}_k^{(k-1)} = \mathbf{X}_k$ and consider just the convergence of the last $u$ rows

$$\mathbf{X}_u^{(k-1)} = \tilde{\mathbf{A}}_{uk}\mathbf{X}_k + \tilde{\mathbf{A}}_{uu}\mathbf{X}_u^{(k-1)}.$$

We can look at the stationary behaviour by unrolling this recursion and taking the limit to find stationary state

$$\lim_{n \to \infty} \mathbf{X}_u^{(n)} = \lim_{n \to \infty} \tilde{\mathbf{A}}_{uu}^n\mathbf{X}_u^{(0)} + \left(\sum_{i=1}^{n} \tilde{\mathbf{A}}_{uu}^{(i-1)}\right)\tilde{\mathbf{A}}_{uk}\mathbf{X}_k.$$

Using Lemma 1 we find $\lim_{n \to \infty} \tilde{\mathbf{A}}_{uu}^n\mathbf{X}_u^{(0)} = \mathbf{0}$ and the geometric series converges giving us the following limit

$$\lim_{n \to \infty} \mathbf{X}_u^{(n)} = \left(\mathbf{I}_u - \tilde{\mathbf{A}}_{uu}\right)^{-1}\tilde{\mathbf{A}}_{uk}\mathbf{X}_k = -\mathbf{\Delta}_{kk}^{-1}\tilde{\mathbf{A}}_{uk}\mathbf{X}_k.$$

□

| Dataset | Nodes | Edges | Features | Classes |
|---|---|---|---|---|
| Cora | 2,485 | 5,069 | 1,433 | 7 |
| CiteSeer | 2,120 | 3,679 | 3,703 | 6 |
| PubMed | 19,717 | 44,324 | 500 | 3 |
| Photo | 7,487 | 119,043 | 745 | 8 |
| Computers | 13,381 | 245,778 | 767 | 10 |
| OGBN-Arxiv | 169,343 | 1,166,243 | 128 | 40 |
| OGBN-Products | 2,449,029 | 123,718,280 | 100 | 47 |

Table 3: Dataset statistics.

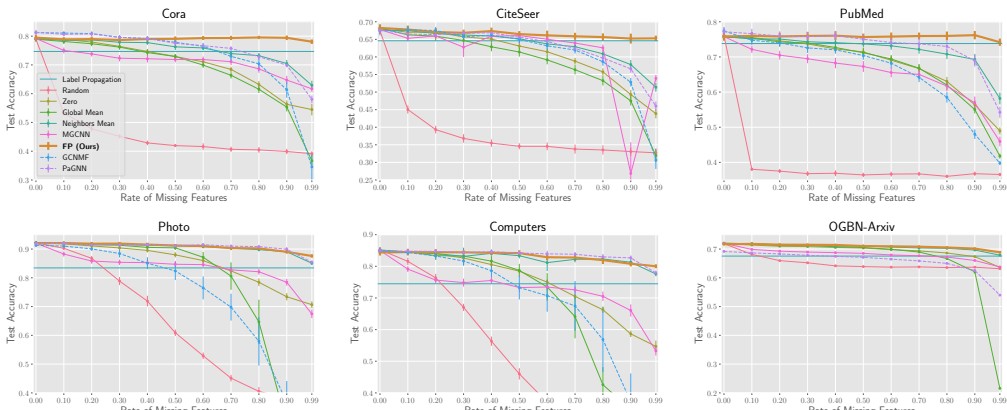

Figure 6: Test accuracy for varying rate of missing features on six common node-classification benchmarks. For methods that require a downstream GNNs, a 2-layer GraphSAGE (Hamilton et al., 2017) is used. On OGBN-Arxiv, GCNMF goes out-of-memory and is not reported.

## A.3 Baselines' Implementation and Tuning

**Label Propagation** We use the label propagation implementation provided in Pytorch-Geometric (Fey & Lenssen, 2019). Since the method is quite sensitive to the value of the $\alpha$ hyperparameter, we perform a gridsearch separately on each dataset over the following values: $[0.1, 0.2, 0.3, 0.4, 0.5, 0.6, 0.7, 0.8, 0.9, 0.95, 0.99]$.

**MGCNN** We re-implement MGCNN (Monti et al., 2017) in Pytorch by taking inspiration from the authors' public TensorFlow code [6]. For simplicity, we use the version of the model with only graph convolutional layers and without an LSTM. For the matrix completion training process, we split the observed features into 50% input data, 40% training targets and 10% validation data. Once the MGCNN model is trained, we feed it the matrix with all the observed features to predict the whole feature matrix. This reconstructed features matrix is then used as input for a downstream GNN (as for the feature-imputation baselines).

## A.4 Discussion Over Baselines' Performance

**Neighborhood Averaging** As for some intuition to why Neighborhood Averaging works so well, let's assume to have a single feature channel for simplicity. The average of neighbors' features is a good estimator of the true feature of a given node when the feature is observed for enough neighbors (and it is homophilous over the graph). However, as the rate of missing features increases, the feature may be present for only a few neighbors (or none at all), causing the estimator to have a much higher variance (and therefore less likely to be correct). On the other hand, Feature Propagation allows information to travel longer distances in the graph by repeatedly multiplying by the diffusion matrix. This means that even if we do not observe the feature for any of a node's neighbors, we can still estimate it from nodes further away in the graph. This can be observed empirically: the gap between Neighborhood Averaging and Feature Propagation becomes increasingly significant for higher rates of missing features.

**Zero vs Random** We thank the Reviewer for bringing up this important point, and we will stress it in the revised version of our paper. Our intuition is that in models such as GCN and GraphSage, where node embeddings are computed as (weighted) average of neighbors embeddings, the effect of the Zero baseline is simply to reduce the norm of the average embeddings of all nodes (since all nodes have the same expected proportion of neighbors with missing features). On the other hand, the Random baseline corrupts this weighted average. More generally, while for a GNN model it could be relatively easy to learn to ignore features set to zero, and only focus on known (non-zero) features, it would be basically impossible for the model to do the same when setting the missing features to a random value.

However, we find Random to perform better than Zero when all features are missing. This is in line with findings in the literature (Sato et al., 2021; Abboud et al., 2021), where Random features have been shown to work well in conjunction with GNNs as they act as signatures for the nodes. On the other hand, if all nodes have all zero vectors, it becomes basically impossible to distinguish them. After applying a GNN, all nodes will still have very similar embeddings and the task performance will be close to a random guess.

---

[6]https://github.com/fmonti/mgcnn

