# OpenReview forum: "On the Unreasonable Effectiveness of Feature Propagation in Learning on Graphs with Missing Node Features"
_ICLR.cc/2022/Conference — ICLR 2022 Submitted_

### Official Review · Reviewer_U1iE · 2021-10-23

**Correctness:** 4
**Technical Novelty And Significance:** 3
**Empirical Novelty And Significance:** 3
**Recommendation:** 8
**Confidence:** 3

**Main Review:**

Strengths:

The paper is well written, and easy to follow. The problem the authors address is interesting, and the solution provided is simple, yet elegant. The experimental results are thorough, both in terms of comparing performance and speed of the method.

Weaknesses:

It's not completely clear how the proposed method extends to vector valued features. Is the process repeated independently for each feature? or is there some "interdependencies" between feature dimensions that can be made use of? How does that affect the computational complexity?

**Summary Of The Paper:**

The authors consider the case of training GNNs with missing features. The proposed framework consists of a diffusion based step prior to training the GNN. A discretization of the approach leads to what they term "feature propagation", a scalable method to impute features on the graph. The assumption is that the energy function that determines how related the features are to each other can be learnt from the data. The authors consider the gradient flow and it's solution to minimize the dirichlet energy, which is far more scalable than computing the closed form solution. Experiments on multiple datasets show that the proposed method outperforms several baselines.

**Summary Of The Review:**


Initial assessment : accept

The paper is well written, and easy to follow. The problem the authors address is interesting, and the solution provided is simple, yet elegant. The experimental results are thorough, both in terms of comparing performance and speed of the method. Missing value imputation is a problem that is often ill-addressed when it comes to ML research. In this paper, the authors specifically look at imputation in the context of training GNNs, and use the graph to perform the imputation.

I feel the authors need to address the difference between the proposed method and label propagation a bit more. They claim it's different because of labels being discrete and the diffusion operator being different, but I'm not clear why that's a deal breaker. Won't the algorithm be the same but with a new operator?

Another point to address is the extension to vector valued features. They claim the extension is straightforward, but a note to this effect will be nice.

other comments:
P1 last paragraph : I’m not sure if being unaware to the task is a bad thing necessarily.  isn't it better to do imputation in a task agnostic fashion?

Fig 3: Please make the text larger. The figures are hard to read.

Fig 3: Also any intuition why Random is so much worse than Zero in all these datasets?

---

> ### Author Response · Authors · 2021-11-13
> **Response to Reviewer U1iE**
>
> We thank the Reviewer for their helpful comments. We address common questions (including the difference with Label Propagation and how our approach works for vector-valued features) in our general response to all the Reviewers. Other specific questions/concerns are addressed below:
>
> > “I’m not sure if being unaware to the task is a bad thing necessarily. isn't it better to do imputation in a task agnostic fashion?”
>
> We agree that being task-agnostic can be advantageous in certain settings (as reconstructing the features only once and then using them for multiple downstream tasks). Feature propagation is indeed task-agnostic. We have rephrased the wording of the paper.
>
>  > “Fig 3: Please make the text larger. The figures are hard to read.”
>
> Thanks for pointing this out. We’ve changed it in the updated version of the paper.
>
> > Fig 3: Also any intuition why Random is so much worse than Zero in all these datasets?
>
> We thank the Reviewer for bringing up this important point, and we will stress it in the revised version of our paper. Our intuition is that in models such as GCN and GraphSage, where node embeddings are computed as a weighted average of neighbors embeddings, the effect of the Zero baseline is simply to reduce the norm of the average embeddings of all nodes (since all nodes have the same expected proportion of neighbors with missing features). On the other hand, the Random baseline corrupts this weighted average. More generally, while for a GNN model it could be relatively easy to learn to ignore features set to zero, and only focus on known (non-zero) features, it would be basically impossible for the model to do the same when setting the missing features to a random value.
>
> However, we find Random to perform better than Zero when all features are missing. This is in line with findings in the literature ([1, 2]), where Random features have been shown to work well in conjunction with GNNs as they act as signatures for the nodes. On the other hand, if all nodes have all zero vectors, it becomes basically impossible to distinguish them. After applying a GNN, all nodes will still have very similar embeddings and the task performance will be close to a random guess.
>
> > “How does (extending the method to vector-valued features) affect the computational complexity?”
>
> The method extends to vector-valued features by simply replacing the feature vector $\mathbf{x}$ with a $n \times d$ feature matrix $\mathbf{X}$ in Algorithm 1, where $n$ is the number of nodes and $d$ the number of features. Regarding the computational complexity, the sparse-to-dense matrix multiplication incurs in a $O(m \times n \times d)$ complexity, where $m$ is the number of non-zero entries in $\mathbf{A}$. In practice, since the operation is independent for each feature channel, the operation is highly parallel and runs extremely fast on a GPU. We clarified this in the updated version of the paper.
>
> [1] Ryoma Sato, Makoto Yamada, and Hisashi Kashima. Random features strengthen graph neural networks. In Proceedings of the 2021 SIAM International Conference on Data Mining, SDM, pp. 333–341. SIAM, 2021.
>
> [2] Ralph Abboud, ˙Ismail ˙Ilkan Ceylan, Martin Grohe, and Thomas Lukasiewicz. The surprising power of graph neural networks with random node initialization. In Proceedings of the Thirtieth International Joint Conference on Artificial Intelligence, IJCAI-21, pp. 2112–2118, 2021.

---

> > ### Comment · Reviewer_U1iE · 2021-11-27
> > **post discussion feedback**
> >
> > I checked out the responses to my review, and the points raised in other reviews. Agreed with the part about the main novelty being more on the preprocessing side, and the authors (maybe (?) ) have addressed that. I'm going to keep my score for now.

---

> ### Author Response · Authors · 2021-11-22
> **Final Revised Version of the Paper Uploaded**
>
> We uploaded a final revised version of the paper, and all changes from the initial version are summarized in the general response to all Reviewers. If the Reviewer feels their concerns have been adequately addressed, we hope they would consider increasing their score accordingly.

---

### Official Review · Reviewer_bVRh · 2021-10-30

**Correctness:** 4
**Technical Novelty And Significance:** 2
**Empirical Novelty And Significance:** 2
**Recommendation:** 5
**Confidence:** 4

**Main Review:**

Through numerical experiments the authors illustrate the benefit of this simple approach. Nonetheless, my main concern has to do with the novelty of this. Using the quadratic form of the graph Laplacian (Dirichlet energy) for label propagation or graph signal interpolation has been used several times over the last decade (as referenced in this paper). Moreover, using a gradient descent approach instead of inverting the Laplacian has also been used quite often. E.g., see the tutorial "Signal processing on higher-order networks: Livin’on the edge... and beyond" where in Section 2.4.2 the authors review some typical formulations of signals on graphs before going into higher-order networks. Notice that Equation (4) contains (I-L)^k, which corresponds to several applications of the (normalized) adjacency matrix. In fact, it is mentioned that "This may be interpreted in terms of k gradient descent steps of the cost function (Dirichlet energy)".

From the perspective of graph signal processing, the true solution with the inverse of the Laplacian can be seen as an IIR filter and this is an FIR approximation of the filter. This idea of graph filter design has also been studied quite a bit over the last 7 years.

**Summary Of The Paper:**

This paper proposes a fast way to do node feature imputation in graph learning settings. In a nutshell, the idea is to consider the graph topology in the imputation by doing a feature by feature minimization of the Laplacian quadratic form. However, since the exact solution to this requires a matrix inversion, this quantity is minimized by a gradient descent procedure that relies on sparse matrix multiplications.


**Summary Of The Review:**

Under this perspective, I regard that the novelty in this paper is limited.

---

> ### Author Response · Authors · 2021-11-13
> **Response to Reviewer bVRh**
>
> We thank the Reviewer for their helpful comments. We address common questions (including the concerns related to novelty and the difference with Label Propagation) in our general response to all the Reviewers. In short, we do not claim that the proposed algorithm based on the minimization of Dirichlet energy is the main novelty of the paper; instead, we believe the main contributions lie in the significance of the problem itself (prediction with missing node features), the solid theoretical motivation of the proposed approach, its robustness to high rates of missing features, and its characteristics of being generic, fast and scalable to extremely large graphs, provide important empirical insights that we believe are explicitly valued by the ICLR community.
>
> Other specific questions/concerns are addressed below:
>
> > “From the perspective of graph signal processing, the true solution with the inverse of the Laplacian can be seen as an IIR filter and this is an FIR approximation of the filter. This idea of graph filter design has also been studied quite a bit over the last 7 years.”
>
> We thank the Reviewer for pointing this connection out. Our method can indeed be interpreted as a polynomial approximation of the rational filter. Although this interpretation generalizes directly from classical filter theory, if the Reviewer provides specific references on this in the graph signal processing literature (or other references related to our work) we will be happy to include and discuss them in the paper.

---

> > ### Comment · Reviewer_bVRh · 2021-11-18
> > **Follow-up**
> >
> > I appreciate your general response and the specific response to my comment. I just want to confirm with you that I am not missing anything here. If we distil the proposed methodology to its core, what is proposed in the paper is to do:
> >
> > 1. Classical label propagation BUT on the feature space (each feature separately) instead of the node classes, to impute missing data.
> > 2. Run a GNN.
> >
> > The "Difference between FP and LP" mentioned lies on the fact that instead of using propagation in the labels (LP) and ignoring all the features, you do propagation in the features (FP) and then employ a GNN that uses those features. In technical terms they are the same procedure, it is just where you apply it and whether you use this as a pre-processing step into a (more sophisticated) classifier.
> >
> > Do you agree with the description above?

---

> > > ### Author Response · Authors · 2021-11-18
> > > **Response to Follow-Up**
> > >
> > > We agree with the description above. As previously stated, FP and LP are conceptually similar: the former propagates the features, while the latter propagates labels (in details, the specifics of the diffusion operators may vary; e.g. since LP propagates class probabilities, their diffusion operator is row-stochastic, while ours is not).
> > >
> > > In practice, however, the difference is substantial, which we believe to be the key empirical contribution of the paper: FP (combined with a GNN) is an extremely effective solution to the problem of missing node features, requiring a very low % of present features to work well, and significantly outperforming LP which instead ignores the features altogether.

---

> ### Author Response · Authors · 2021-11-22
> **Final Revised Version of the Paper Uploaded**
>
> We uploaded a final revised version of the paper, and all changes from the initial version are summarized in the general response to all Reviewers. If the Reviewer feels their concerns have been adequately addressed, we hope they would consider increasing their score accordingly.

---

### Official Review · Reviewer_JE1t · 2021-11-02

**Correctness:** 4
**Technical Novelty And Significance:** 2
**Empirical Novelty And Significance:** 2
**Recommendation:** 5
**Confidence:** 4

**Main Review:**

The strengths:

1. The paper is clearly written and easy to follow
2. The performance on missing 99% of the features is impressive
3. The speed improvement is promising

The weaknesses:

1. FP is not a new idea/algorithm, it is essentially same as label propagation (LP) by Zhu et al. [1]. Although the authors argued that "Differently from our setting of diffusion of continuous node features, they deal with discrete label classes directly, resulting in a different diffusion operator.", from my understanding, there is not substantial difference. LP also propagate the label probability, which is continuous. The derivation, the update formula of FP are the same as LP, the only difference is to substitute the weight matrix in LP with Laplacian matrix. Interpreting label propagation via Dirchlet energy is also proposed in [2], where instead of using $x_i$, $x_j$, Solomon et al. were using $f_i$, $f_j$, which are label function of node $i$ and $j$. Consider $x_i$'s as the soft label with label function $f_i$, then they are essentially the same thing.
2. The experiment part shows that simple averaging of the neighbors gives good result when the missing portion is $\leq$ 50%. In that case, I think it is also important to show experimental results of algorithms that do not use node features, to see if there is really a gap on the performance. (Note that under strong assumption of homophily, it is not surprising that simple averaging of neighbors works well. FP or LP takes account the topology of the graph and changes simple averaging of neighbors to weighted average of the whole network. The global information of the graph is the key to a better performance in this case, and I doubt other algorithms using the global information can also get similar good performance.) For example, one can use simple label propagation on the node classes, or using LINE,
DeepWalk or node2vec to get a embedding of each node and then classifying with the embeddings. If the performance is similar as what FP gives when there are 99% features missing, I would argue that in that case we can simply go with classical algorithms using only the network topology. We will only use GNN when we have enough features available, while in which case it seems simple averaging of neighbors is good enough. The paper is not so convincing at the current stage.

[1] Xiaojin, Zhu, and Ghahramani Zoubin. "Learning from labeled and unlabeled data with label propagation." Tech. Rep., Technical Report CMU-CALD-02–107, Carnegie Mellon University (2002).

[2] Solomon, Justin, Raif Rustamov, Leonidas Guibas, and Adrian Butscher. "Wasserstein propagation for semi-supervised learning." In International Conference on Machine Learning, pp. 306-314. PMLR, 2014.

**Summary Of The Paper:**

The paper address the problem of  missing node features in graph by conducting feature propagation (FP). An iterative updated algorithm is presented. FP is a preprocessing step and can be used in multiple downstream tasks. Experiments on node classification shows the effectiveness of FP. The authors also provided comparison on running time and showed how homophily affects the performance of FP.

**Summary Of The Review:**

In general I like the idea of using FP as a preprocessing step. However, I am not convinced that the idea is novel enough to be accepted in ICLR, details are referred to my main review.

---

> ### Author Response · Authors · 2021-11-13
> **Response to Reviewer JE1t**
>
> We thank the Reviewer for their helpful comments. We address common questions (including the difference from Label Propagation and concerns regarding the novelty of the paper) in our general response to all the Reviewers. Other specific questions/concerns are addressed below:
>
> > “It is also important to show experimental results of algorithms that do not use node features, to see if there is really a gap on the performance.”
>
> We agree that this is an important experiment to make sure that there is an advantage in reconstructing the missing features over just ignoring the features by using a feature-agnostic method. As suggested by the Reviewer, we added a comparison with Label Propagation (on the class labels) on all datasets. We used the implementation provided by PyTorch Geometric and tuned the $\alpha$ hyperparameter on the validation set.
> Figure 3 in the revision of the paper presents the results. In all datasets, even in the extreme case of 99% missing features, Feature Propagation + GCN outperforms Label Propagation. On some datasets, such as Cora, Photo, and Computers, the gap is especially large. We conclude that reconstructing the missing features is indeed useful, even in cases of high rates of missing features, as it consistently outperforms feature-agnostic methods.
>
> | Dataset      | FP (99% Missing) | Label Propagation |
> | ----------- | ----------- | ----------- |
> | Cora       | 78.22\% |               74.68\% |
> |  CiteSeer     | 65.40\% |               64.59\% |
> |    PubMed    | 74.29\% |               73.80\% |
> |     Photo       | 87.73\% |               83.44\% |
> | Computers  | 80.94\% |               74.47\% |
> |OGBN-Arxiv | 69.09\% |               67.55\% |

---

> ### Author Response · Authors · 2021-11-22
> **Final Revised Version of the Paper Uploaded**
>
> We uploaded a final revised version of the paper, and all changes from the initial version are summarized in the general response to all Reviewers. If the Reviewer feels their concerns have been adequately addressed, we hope they would consider increasing their score accordingly.

---

> ### Comment · Reviewer_JE1t · 2021-12-01
> **Post Rebuttal**
>
> I have read the rebuttal/revised paper and other reviews, my concerns are not fully addressed and I decide to keep my score.

---

### Official Review · Reviewer_AjnH · 2021-11-03

**Correctness:** 3
**Technical Novelty And Significance:** 3
**Empirical Novelty And Significance:** 3
**Recommendation:** 5
**Confidence:** 4

**Details Of Ethics Concerns:**

As mentioned in the paper, these approaches, for matrix completion, can be applied to problems that can have a negative impact. Matrix completion approaches are applied to recommender systems, and as they blindly recommend items that a user will likely rate highly given previous ratings they tend to put users into their own bubble of information, that can unfortunately reinforce false beliefs (worsening the misinformation problems in society), additionally it creates divides between groups of people that have similar ratings, and on large platforms for information this can have negative effects for society (larger separation of society). However, I think this impact is beyond the scope of this paper, but hopefully future work can improve the evaluation metrics of these approaches to consider this potentially negative impact to society. I think it is worth being aware of this though.

**Main Review:**

This paper addresses a very interesting problem that is relevant to real-world applications; dealing with missing node features for graph neural networks (GNNs). The approach is based on minimising Dirichlet energy, which is a emthod previously used for graph regularisation and has relations with graph Laplcians, so is well suited to working with data on graph topology. The author(s) derive an efficient iterative algorithm which makes the approach scalable to large graphs which makes it more practical for real-world applications. Many public datasets were used for benchmarking that show very promising results.

In the experiments, it's surprising how well the nearest-neigbour averaging worked. In the comparison method's own papers SAT paper has a 'NeighAggre' baseline that seems to do something similar but gets worse scores than their method and GCNMF compares with kNN with k=5, and this also performs worse than their method. Do you have some insights into why neighbourhood averaging performs so well?

This method is a pre-processing step, not integrated into the GNN, as such many other methods for estimating the missing values could be used. One justification of this choice of approach is that is acts as a low-pass filter, and the GNN mainly learns from the lower frequency components.  This work is very close to matrix completion with graph side information.In this domain there are other methods that are highly scalable and are able to accurately estimate missing values when large amounts of the data (>99%) are missing [1, 2, 4]. Furthermore, one method uses a similar optimisation approach of iterative dense-sparse matrix mulitplication [1]. Worth note is that an extension to [1] will also remove edges in the graph that do not have high-homophily that could improve results for low-himopholy results. Also [3] is a kernel method for matrix completion that allows for a choice of covariance functions, including a diffusion kernel that is therefore similar to this approach. I think it is worth at mentioning this in comparison to this approach and even adding emprical evaluation against these methods, especially [1].

Other small comments:

You report using V100 GPUs (16 or 32Gb?), how much memory did GCNMF take before ran out of memory.

It wasn't crystal clear how the approach works for vector node attributes, do you simply repeat the algorithm for each dimension separately?

"Proposition 1: inverse of Laplacian depends on a connected graph, as eigen values are 0 if graph has clusters." How true is this assumption in real graphs, and what can be done in the case where a graph is not connected?

How do you know 40 iterations is enough, is there a easy guide to this and what happens if you keep running the algorithm, will it eventually make all missing featues almost the same if run for a large number of iterations?

"For a fair comparison, we use the same standard hyperparameters for all methods across all experiments." Does this mean that the downstream 2-layer GCN model has the same hyperparameters?

Edits:
    A_u should be A_uu ?

    Zhu and Ghahramani reference is incomplete

    donwstream task -> downstream task


1. Rao, Nikhil, et al. "Collaborative Filtering with Graph Information: Consistency and Scalable Methods." NIPS. Vol. 2. No. 4. 2015.

2. Strahl, Jonathan, et al. "Scalable probabilistic matrix factorization with graph-based priors." Proceedings of the AAAI Conference on Artificial Intelligence. Vol. 34. No. 04. 2020.

3. Zhou, Tinghui, et al. "Kernelized probabilistic matrix factorization: Exploiting graphs and side information." Proceedings of the 2012 SIAM international Conference on Data mining. Society for Industrial and Applied Mathematics, 2012.

4. Monti, Federico, Michael M. Bronstein, and Xavier Bresson. "Geometric matrix completion with recurrent multi-graph neural networks." arXiv preprint arXiv:1704.06803 (2017).



**Summary Of The Paper:**

A graph neural network (GNN) pre-processing step for completing missing node features is presented: feature propagation (FP). The method is based on minimisation of Dirichlet energy. A more efficient iterative update aproach is derived avoiding matrix inversion (cubic operation) to a series of sparse-to-dense matrix multiplications. The method acts similar to a low-pass filter for missing features, which complements the GNN learning algorithm. Relevant work is covered and similarities and differences mentioned in many cases. Experiments on seven publicly available datasets show that the method is effective and performs especially well when a large proportion of the feature values are missing (above 90%). A run-time analysis of wall-clock performance compares their method to two state-of-the-art methods for handling missing node features in GNNs and their method is 3 x faster in these comparison. A weakness of their method, FP, is analysed; FP assumes the graphs are homophlic, that nearby nodes have similar node features but if this is not the case then the method does not perform well. This is some interesting future work.

**Summary Of The Review:**

As this is a preprocessing step that essentially is similar to matrix completion with graph side information, then I believe that comparison with this literature is missing. Otherwise this is a very good paper. The approach they have for the problem, which could be seen as matrix completion with graph side information, is I believe novel in that domain, as well as in this framework for graph neural networks. Therefore, comparison with this domain is important here, as other methods in this domain could also be well justified as pre-processing steps. One such method that both scalable (using similar final equation, but dervied from a different approach, conjugate gradients), must be compared both in the paper and experiments in my opinion. Otherwise, it feels like the work is trying to avoid comparison with similar methods by claiming this work is related to GNN, while really it feels like this is a matrix completion with graph side information method, that is then used for GNNs.

After feedback:

I still feel that the motivation for using this particular method of estimating the missing values is not clearly better than other approaches. Also, the link to GNN seems weak as essentially this is a method for completing missing data and with this seems better suited to the domain for estimating missing values, where a stronger comparison with other missing values estimating methods can be compared - for example in comparison to recommender system approaches like collaborative filtering with graph side information. The link to GNN could be a small subsection in that style of paper. I don't see why running GNN is such a large part of this work when it's a stand alone preprocessing step. Therefore I've lowered my score from 6 to 5.

---

> ### Author Response · Authors · 2021-11-13
> **Response to Reviewer AjnH**
>
> We thank the Reviewer for their helpful comments. We address common questions (including how our approach works for vector-valued features) in our general response to all the Reviewers. Other specific questions/concerns are addressed below:
>
> > “ I think it is worth at mentioning this in comparison to this (matrix completion) approach and even adding emprical evaluation against these methods, especially [1].”
>
> We agree with the Reviewer that a comparison with matrix completion approaches would be extremely informative, and for pointing us to a scalable matrix completion approach. We are currently working on re-implementing [1] and we hope to be able to share the results of the comparison early next week.
>
> > “Do you have some insights into why neighbourhood averaging performs so well?”
>
> We thank the Reviewer for bringing up this important point, and which we comment on in the revised version of our paper. Neighborhood Averaging can be seen as a first-order approximation of Feature Propagation, where only one step of propagation is performed (and with a slightly different normalization of the diffusion operator).
> As for some intuition as to why Neighborhood Averaging works so well, let’s assume to have a single feature channel for simplicity. The average of neighbors’ features is a good estimator of the true feature of a given node when the feature is observed for enough neighbors (and it is homophilous over the graph). However, as the rate of missing features increases, the feature may be present for only a few neighbors (or none at all), causing the estimator to have a much higher variance (and therefore less likely to be correct). On the other hand, Feature Propagation allows information to travel longer distances in the graph by repeatedly multiplying by the diffusion matrix. This means that even if we do not observe the feature for any of a node’s neighbors, we can still estimate it from nodes further away in the graph.
> This can be observed empirically: the gap between Neighborhood Averaging and Feature Propagation becomes increasingly significant for higher rates of missing features.
> > “You report using V100 GPUs (16 or 32Gb?), how much memory did GCNMF take before ran out of memory.”
>
> Each V100 GPU has 16GB of memory, which was not sufficient for GCNMF to run on OGBN-Arxiv.
> > "Proposition 1: inverse of Laplacian depends on a connected graph, as eigen values are 0 if graph has clusters." How true is this assumption in real graphs, and what can be done in the case where a graph is not connected?
>
> Our analysis and method apply in this case straightforwardly: if a graph is disconnected, we can apply Feature Propagation to each connected component independently (it is possible to order the nodes in such a way that all the matrices have a block-diagonal structure, with each block corresponding to a connected component of the graph). This is analogous to message passing convolutional layers: when dealing with a graph with multiple connected components, they operate on each component independently (i.e. there is no message/propagation between components).
> > “How do you know 40 iterations is enough, is there a easy guide to this and what happens if you keep running the algorithm, will it eventually make all missing featues almost the same if run for a large number of iterations?”
>
> While simply multiplying the features by a diffusion matrix would lead to oversmoothing (as it commonly happens in graph neural networks), in Feature Propagation the known features are reset to their true value after each diffusion step (in PDE terminology, these are the boundary conditions of the diffusion equation). This prevents oversmoothing and guarantees that the solution at convergence is meaningful. We empirically found that on all datasets we experimented on, 40 iterations were enough to reach convergence (i.e. the predicted missing features do not change anymore). However, it is possible to also just check for convergence after each step, which does not require knowing the number of iterations in advance.
> > "For a fair comparison, we use the same standard hyperparameters for all methods across all experiments." Does this mean that the downstream 2-layer GCN model has the same hyperparameters?
>
> Yes, the downstream 2-layer GCN model has the same hyperparameters in all experiments. Moreover, the same hyperparameters are also used for GCNMF and PaGNN to ensure a fair comparison.
> > “Edits: A_u should be A_uu ?”
> “Zhu and Ghahramani reference is incomplete”
> “donwstream task -> downstream task”
>
> Thanks for pointing these out, we have corrected them in the paper.
>
> [1] Rao, Nikhil, et al. "Collaborative Filtering with Graph Information: Consistency and Scalable Methods." NIPS. Vol. 2. No. 4. 2015.

---

> > ### Comment · Reviewer_AjnH · 2021-11-21
> > **Implementation of Rao et al 2015**
> >
> > You should not need to implement Rao et al 2015 [1] yourself, as a fast implementation is available online:
> >
> > https://github.com/rofuyu/exp-grmf-nips15

---

> > > ### Author Response · Authors · 2021-11-22
> > > **Comparison with Matrix Completion**
> > >
> > > We thank the Reviewer for pointing us to the public implementation for [2]. In the end, we opted for comparing to the other matrix completion method they suggested, MGCNN [1], since it is a more recent (2017 vs 2015) and more cited work and a public *python* implementation is available online (while the code for [2] is in C++), which made it easier for us to incorporate it in our pipeline to run all necessary experiments (which amount to over a thousand runs for each method). We also cited both methods in Section 4.
> > >
> > > The updated version of the paper contains the new results in Figures 3 and 6. Overall, FP outperforms MGCNN, with particularly large gaps for high rates of missing features (>90%).
> > >
> > > [1] Monti, Federico, Michael M. Bronstein, and Xavier Bresson. "Geometric matrix completion with recurrent multi-graph neural networks." arXiv preprint arXiv:1704.06803 (2017).
> > >
> > > [2] Rao, Nikhil, et al. "Collaborative Filtering with Graph Information: Consistency and Scalable Methods." NIPS. Vol. 2. No. 4. 2015.

---

> > > > ### Comment · Reviewer_AjnH · 2021-11-26
> > > > **Comparison with Matrix Completion with Graph Side Information**
> > > >
> > > > Note that Monti et al. 2017 is not a scalable approach and the experiments are done on 3k by 3k subsets of the truly large datasets, so not a good example of highly scalable methods. Rao et al. 2015 is highly scalable and performs exceptionally well. The publicly available code is C++ but as part of MATLAB code, so it can easily be run in MATLAB. The high citation count for Monti et al. 2017 can be largely attributed to it's novel approach, which is very interesting and applies geometric deep learning.

---

> > ### Author Response · Authors · 2021-11-22
> > **Response to Reviewer AjnH (Cont'd)**
> >
> > > "As mentioned in the paper, these approaches, for matrix completion, can be applied to problems that can have a negative impact. [...] However, I think this impact is beyond the scope of this paper, but hopefully future work can improve the evaluation metrics of these approaches to consider this potentially negative impact to society. I think it is worth being aware of this though."
> >
> > We completely agree with the Reviewer about the potential negative impact of matrix completion approaches, and that obtaining a deeper understanding of this impact (and how to mitigate it) is an extremely interesting and important research direction, albeit outside the scope of this paper. We will consider adding a comment on this in the revised version of the paper.

---

> ### Author Response · Authors · 2021-11-22
> **Final Revised Version of the Paper Uploaded**
>
> We uploaded a final revised version of the paper, and all changes from the initial version are summarized in the general response to all Reviewers. If the Reviewer feels their concerns have been adequately addressed, we hope they would consider increasing their score accordingly.

---

> ### Author Response · Authors · 2021-11-30
> **Response to Updated Review after Initial Feedback**
>
> > "I still feel that the motivation for using this particular method of estimating the missing values is not clearly better than other approaches. Also, the link to GNN seems weak as essentially this is a method for completing missing data and with this seems better suited to the domain for estimating missing values, where a stronger comparison with other missing values estimating methods can be compared - for example in comparison to recommender system approaches like collaborative filtering with graph side information. The link to GNN could be a small subsection in that style of paper. I don't see why running GNN is such a large part of this work when it's a stand alone preprocessing step. Therefore I've lowered my score from 6 to 5."
>
> As mentioned in the paper, the goal of our work is to solve the problem of missing features in the context of some downstream tasks (eg. node classification). This is in contrast to classical matrix completion works, where the goal is to reconstruct the features themselves as well as possible. In our scenario, we do not really care about how well the features are reconstructed, but rather that the reconstructed features work well as input to a GNN to solve downstream tasks. Nevertheless, we compare with a matrix completion approach and show that FP significantly outperforms it on all benchmarks on the node-classification task (which again, is what we are interested in). We further explain in the paper that the reason why FP works so well in conjunction with GNNs is that they have similar assumptions (homophily) and use a similar mechanism (diffusion). As a result, FP focuses on reconstructing the component of the original features which are actually used by the downstream GNN (the low-frequency component of the features).
> With this explanation of motivation we hope the reviewer will raise their score again.

---

### Author Response · Authors · 2021-11-13
**General Response to all Reviewers**

We thank the Reviewers for their thoughtful comments! We are delighted to hear that reviewers find the problem we tackle to be relevant and interesting (AjnH, bVRh), our approach to be simple, practical while also elegant (AjnH, bVRh), the experimental results to be impressive (AjnH, JE1t, bVRh) and the paper to be clearly written and easy to follow (JE1t, bVRh).

We address the most important/common questions here while answering specific questions in our individual responses to each Reviewer.


**Novelty**

We do not claim to have invented minimization of the Dirichlet energy, which is a standard notion known for decades, if not a century, in mathematical physics and differential geometry. In fact, this idea predates Label Propagation papers and has been explored e.g. in image processing starting from Perona-Malik diffusion in 1990. Label Propagation itself is an application of this concept.

However, the application of this approach to the problem of learning with missing features has never been presented before. Referring to ICLR’s Review Guidelines, we believe our paper presents “empirical insights” novelty rather than “technical contribution” novelty. Our paper stretches the limits of what has so far been known about the performance of GNNs with missing features, and the fact that a very simple and efficient method can outperform much more complex recent methods is a significant insight for the community and has an immediate practical application (see “Significance of the Problem” paragraph below). We tried to convey this empirical contribution even in the title (“unreasonable effectiveness”).


**Significance of the Problem**

The problem of missing node features is extremely widespread and relevant in the real world but has barely been considered before. GNN models typically assume a fully observed feature matrix. However, in many important real-world scenarios, including the largest applications of GNNs to date, this is not the case. For example, using GNNs to make predictions about users in social networks would ideally leverage optional user input such as date of birth or location. Modern e-commerce recommendation systems (which use the bipartite user-product graph) will typically utilize product metadata, which is usually manually completed by suppliers and often has missing fields. Systems to predict user returns will similarly rely on an optional customer survey that is frequently not completed. More generally, with the rising awareness around digital privacy, user data is increasingly available only upon explicit user consent.  In all the above cases, the feature matrix contains missing values and most existing GNN models cannot be directly applied.


**Differences from Label Propagation**

While FP and LP are algorithmically similar, the key difference between them lies in how they are used. LP is used to directly perform node classification, taking into account only the graph structure and being unable to make use of node features. On the other hand, FP is used to reconstruct missing features to then feed into a downstream GNN classifier. FP allows a GNN model to effectively combine features and graph structures, even when most of the features are missing. In the updated version of the paper, we include experiments showing that FP+GNN always outperforms LP, also with extremely high rates of missing features.


**Extension of the Method to Vector-Valued Features**

The method extends to vector-valued features by simply replacing the feature vector $\mathbf{x}$ with a $n \times d$ feature matrix $\mathbf{X}$ in Algorithm 1, where $n$ is the number of nodes and $d$ the number of features. Multiplying the diffusion matrix $\mathbf{A}$ by the feature matrix $\mathbf{X}$ diffuses each feature channel independently.


**Contributions**

Finally, we would like to highlight the contributions of our work:
- *Significance of the problem*: missing node features is a relevant and widespread but largely unexplored issue in GNNs.
- *Theoretically motivated*: FP emerges naturally as minimization of the Dirichlet energy
- *Robust to high rates of missing features*: FP can withstand surprisingly high rates of missing features. It was previously unknown that with only 1% of features it was still possible to obtain competitive performance.
- *Generic*: FP can be combined with any GNN model to solve the downstream task; in contrast, GCNMF and PaGNN are specific GCN-type models.
- *Fast and Scalable*: FP takes only around 10 seconds for the reconstruction step on OGBN-Products (a graph with ∼2.5M nodes and ∼123M edges) on a single GPU. GCNMF and PaGNN run out-of-memory on this dataset.

---

### Author Response · Authors · 2021-11-15
**List of All Changes**

We uploaded a revised version of the paper which incorporates the Reviewers' feedback, and we summarize the main changes below:
- *Section 3*: Elaborated on the difference between Feature Propagation and Label Propagation
- *Section 3*: Elaborated on how to extend Feature Propagation to vector-valued features
- *Section 3*: Explained our Feature Propagation works on graphs with multiple connected components
- *Section 4*: We add citations to [1] and [2] as they are relevant graph matrix completion approaches.
- *Section 5*: Compared to Label Propagation (on label classes) as a feature-agnostic baseline. Results are in Figure 3 and Table 2.
- *Section 5*: Compared to MGCNN [1] as a matrix completion baseline. Results are in Figure 3.
- *Section 5*: Used 3 layers and a hidden dimension of 256 for OGBN-Arxiv and OGBN-Products. For OGBN-Arxiv we also used the Jumping Knowledge scheme. This improves the performance of GCN on these dataset significantly (with or without missing features). We described this in the experimental setup paragraph, and the updated results are in Figure 3 and Table 1 and 2.
- *Section A3*: Elaborated on the performance of the feature-imputation baselines, in particular Neighbor Mean and Zero vs Random
- Made the text of all figures larger so that they are easier to read

[1] Monti, Federico, Michael M. Bronstein, and Xavier Bresson. "Geometric matrix completion with recurrent multi-graph neural networks." arXiv preprint arXiv:1704.06803 (2017).

---

### Decision · Program_Chairs · 2022-01-20

**Decision:**

Reject

**Comment:**

This paper presents a Feature Propagation (FP) method for dealing with missing features in graph learning tasks. The FP method is based on minimization of the Dirichlet energy and leads to a diffusion-type differential equation on the graph. Empirical results demonstrated the effectiveness. However, after rebuttal major concerns still remain on the novelty and siginificance, in particular, the connection with label propogation should be better elaborated, which is crucial to understand the contributions of this paper. Considering that, I can't recommend accept the current manuscript. The authors are encouraged to further improve for a more solid publication in the future.